# The Reserve/Maximum Capacity of Melatonin’s Synthetic Function for the Potential Dimorphism of Melatonin Production and Its Biological Significance in Mammals

**DOI:** 10.3390/molecules26237302

**Published:** 2021-12-02

**Authors:** Dun-Xian Tan, Rüdiger Hardeland

**Affiliations:** 1Department of Cell Systems and Anatomy, UT Health San Antonio, San Antonio, TX 78229, USA; 2Johann Friedrich Blumenbach Institute of Zoology and Anthropology, University of Göttingen, 37073 Göttingen, Germany; rhardel@gwdg.de

**Keywords:** melatonin, dimorphism, *ASMT*, *ASMTL*, mitochondria, sex chromosomes, reserved capacity

## Abstract

In this article, we attempt to classify a potential dimorphism of melatonin production. Thus, a new concept of “reserve or maximum capacity of melatonin synthetic function” is introduced to explain the subtle dimorphism of melatonin production in mammals. Considering *ASMT*/*ASMTL* genes in the pseudoautosomal region of sex chromosomes with high prevalence of mutation in males, as well as the sex bias of the mitochondria in which melatonin is synthesized, we hypothesize the existence of a dimorphism in melatonin production to favor females, which are assumed to possess a higher reserve capacity for melatonin synthesis than males. Under physiological conditions, this subtle dimorphism is masked by the fact that cells or tissues only need baseline melatonin production, which can be accomplished without exploiting the full potential of melatonin’s synthetic capacity. This capacity is believed to exceed the already remarkable nocturnal increase as observed within the circadian cycle. However, during aging or under stressful conditions, the reserve capacity of melatonin’s synthetic function is required to be activated to produce sufficiently high levels of melatonin for protective purposes. Females seem to possess a higher reserve/maximum capacity for producing more melatonin than males. Thus, this dimorphism of melatonin production becomes manifest and detectable under these conditions. The biological significance of the reserve/maximum capacity of melatonin’s synthetic function is to improve the recovery rate of organisms from injury, to increase resistance to pathogen infection, and even to enhance their chances of survival by maximizing melatonin production under stressful conditions. The higher reserve/maximum capacity of melatonin synthesis in females may also contribute to the dimorphism in longevity, favoring females in mammals.

## 1. Introduction

Melatonin, a small-molecular-weight antioxidant, is ubiquitously present among organisms, including the domains of bacteria and eukaryotes, and probably also the domain of archaea [1]. Its presence can be traced back to various bacteria, as especially well-documented in some photoautotrophic species, which can use solar energy to synthesize organic substances. Examples are found among species with anoxygenic (*Rhodospirillum rubrum*) and oxygenic (cyanobacteria) photosynthesis [2,3]. UV radiation and oxygen, as well as oxygen derivatives, are quite toxic and can cause oxidative damage to these organisms. Melatonin was probably selected to serve as their first-line antioxidant for their survival [4,5,6]. Due to its specific structure, melatonin is a stable molecule with reduced power and ready to donate one or more electrons to oxidized substances. The oxidation potential of melatonin is approximately +570 mV [7]. This indicates that, at this oxidation potential, melatonin can donate an electron to reactive oxygen species (ROS), which have a much higher oxidation potential than melatonin. Interestingly, the melatonin metabolite, N^1^-acetyl-N^2^-formyl-5-methoxykynuramine (AFMK), which is partially generated with the interaction of melatonin with ROS, exhibits two oxidation potentials, at 456 and 668 mV, respectively [8]. This property indicates that AFMK, as well as its metabolite N^1^-acetyl-5-methoxykynuramine (AMK), can also donate two electrons at the respective oxidation potentials to detoxify ROS [9,10]. The powerful antioxidant capacity of melatonin with its metabolites renders its structure being preserved without any change during evolution [11]. In addition to its antioxidant activity, melatonin performs many functions, including circadian rhythm adjustment, immune-enhancement, reproductive regulation, sleep promotion, and anti-inflammatory activity in animals. All of these functions are acquired at a different stage of evolution [12]. For example, its function in circadian rhythm regulation may have been multiply acquired in different phylogenetic lines, in unicellular organisms, such as dinoflagellates [13], and in metazoans. Its regulatory function in locomotor activity may have originated in diel vertical migration (DVM), which is exhibited by numerous aquatic organisms and is often circadian-controlled [14,15]. In a marine dinoflagellate, the descent to deeper water layers is associated with strong increases in melatonin [16]. It is well documented in many multicellular organisms [14,15], for example, in the annelid worm, *Platynereis dumerilii* [17]. In animals, particularly in vertebrates, the pineal gland has evolved as a specific organ to generate melatonin as the signal of darkness, which synchronizes their physiological rhythmic activities [18]. This signal is extremely important for the reproductive activity of photoperiodic animals. Disturbance of this melatonin signal by artificial light at night or pinealectomy will jeopardize their normal reproductive physiology, either reducing the fertility or giving birth at a wrong season which leads to low survival rate of their offspring [19,20]. Other organs, tissues and cells can also synthesize melatonin [21]. In mammals, extrapineal melatonin does not serve as the signal of darkness, but displays locally functional roles as autocoid, paracoid, or tissue factor [22]. For example, gut-generated melatonin contributes to the preservation of an intact intestinal mucosa and the normal distribution of the gut microbiota [23]. Skin-generated melatonin can protect skin against the UV radiation-induced DNA damage [24]. Embryo-generated melatonin plays a critical role in embryonal development [25]. Due to its pleiotropic activities, the mechanisms of melatonin in a variety of contexts have been extensively studied in test tubes, cells, and animals and are well documented. However, whether the levels of this molecule, as well as its synthetic capacity, exhibit a gender difference is not yet fully clarified. Here, we discuss this important issue from different angles based on the currently available information and our speculations. We hope that this discussion will stimulate further research in this or related areas.

## 2. Melatonin’s Synthetic Pathway in Animals

Melatonin’s synthetic pathways are fairly complicated in different species. For example, bacteria, fungi, plants, and animals share some synthetic reactions but feature others that are completely different, occurring in the melatonin synthetic pathways [26]. These differences include the starting materials, enzymes involved, regulatory factors, and rate-limiting steps, respectively. Here, our focus is given to melatonin synthesis in animals, particularly in vertebrates. In this animal pathway, an amino acid, tryptophan, is the starting material. Due to the absence of the shikimic acid synthetic pathway, most animals are unable to synthesize tryptophan and need to take up this amino acid from their diet. A small number of exceptions in basal metazoans do not necessarily indicate an early loss in animal evolution, since the presence of the respective genes has been shown to be caused by horizontal gene transfer from bacteria [27]. Thus, the availability or blood concentration of tryptophan limits the physiological level of melatonin in animals. This is not the case in microorganisms, fungi and plants, which can synthesize tryptophan depending on their requirement. This is the reason why these species typically feature much higher physiological melatonin levels than animals. Tryptophan is first hydroxylated to form 5-hydroxytryptophan. This step is catalyzed by tryptophan hydroxylase (TPH). Two *TPH* genes (*TPH1* and *TPH2*) are present in animals [28]. TPH1 and TPH2 share similar functions and generate identical products. However, their expression levels are different in various tissues. *TPH2* is mainly expressed in the brain stem and *TPH1* is widely expressed in the pineal gland and other organs and tissues. Even though the expression of *TPH1* exhibits a circadian rhythm with some correspondence to melatonin production, it is not considered the rate-limiting enzyme for melatonin synthesis in animals. Based on the evolutionary history of *TPH1*, this gene has been shown to have been under positive selection for reproductive rhythmicity in photoperiodic mammals, as related to their melatonin production [29]. The second step is the 5-hydroxytryptophan decarboxylation to form 5-hydroxytryptamine (serotonin). This reaction is catalyzed by the enzyme referred to as aromatic amino acid decarboxylase (AADC). Next, serotonin is acetylated to N-acetyl-5-hydroxytryptamine (N-acetylserotonin) by an enzyme referred to as serotonin N-acetyltransferase (SNAT), or arylalkylamine N-acetyltransferase (AANAT), with acetyl-CoA as a cofactor. SNAT/AANAT is considered the rate-limiting enzyme for melatonin synthesis due to the fact that the circadian rhythm of its expression or activity in the pineal gland perfectly matches the serum melatonin circadian rhythm in most of the tested animals. In mammals, it seems that several other non-specific N-acetyltransferases, including arylamine N-acetyltransferase 1 and 2 (NAT1, NAT2) also contribute to serotonin acetylation and melatonin synthesis [30]. Thus, even with *SNAT* deficiency, the animal still can synthesize considerable level of melatonin. For example, in C57BL/6 mice, which have a naturally truncated SNAT without catalytic activity, melatonin can still be detected in their skin tissue and blood with reduced levels [31,32]. The final step is N-acetylserotonin O-methylation to form melatonin catalyzed by N-acetylserotonin-O-methyltransferase (ASMT), also known as hydroxyindole-O-methyltransferase (HIOMT), which uses S-adenosylmethionine (SAM) as a cofactor. It also acts as a rate-limiting step for melatonin synthesis. Evidence has shown that ASMT may play a more important role than SNAT in the regulation of melatonin production under certain conditions, particularly in hamsters [33,34]. Currently, no additional enzyme has been identified that participates in the O-methylation of N-acetylserotonin to form melatonin in mammals. This observation strengthens the argument that ASMT is the most effective and specific rate-limiting enzyme for melatonin synthesis, at least at high production rates [35]. In addition, an alternative pathway has been speculated to exist by Tan et al. [36]. Th authors suggest that serotonin may be first O-methylated by ASMT to form 5-methoxytryptamine (5-methoxyserotonin), since the affinity of ASMT to serotonin is higher than to N-acetylserotonin. The enzyme reaction favors the interaction of ASMT with serotonin, forming 5-methyoxytryptamine. In this alternative pathway, the final step involves the acetylation of 5-methyoxytryptamine by SNAT or NAT1/2 to form melatonin. The detailed pathways are summarized in Figure 1.

This pathway occurs inside mitochondria. The concentrations of Acetyl-CoA and SAM are stable in mitochondria and fit to the K_m_ of SNAT and ASMT, respectively. TPH: tryptophan hydroxylase, AADC: aromatic amino acid decarboxylase, SNAT: serotonin N-acetyltransferase, AANAT: arylalkylamine N-acetyltransferase, ASMT: N-acetylserotonin-O-methyltransferase, HIOMT: hydroxyindole-O-methyltransferase, SAM: S-adenosylmethionine, SAH: S-Adenosylhomocysteine.

## 3. Sites of Melatonin Synthesis

Previously, melatonin was believed to be synthesized in the cytosol of cells, even though there is a lack of well-designed studies to support this notion. After the discovery that primitive bacteria, such as *Rhodospirillum rubrum*, possess the capacity to synthesize melatonin, Tan et al. [37] hypothesized that mitochondria, but not the cytosol, are the major sites for melatonin synthesis. This hypothesis is based on the theory of the endosymbiotic genesis of mitochondria [38]. The group of alphaproteobacteria to which *Rhodospirillum rubrum* belongs is believed to be the precursor of mitochondria [39]. If this is so, then these organelles may still preserve melatonin’s synthetic function inherited from the bacteria. Several studies support the veracity of this hypothesis. Initially, He et al. [40] found that incubation with serotonin leads to melatonin production in a dose-responsive manner in mitochondria isolated from oocytes. These results provided direct evidence that mitochondria possess the capacity to synthesize melatonin. More recently, the SNAT protein was found to exclusively localize in the mitochondria of plants [41] and choroid plexus cells of rats [42]. Suofu et al. [43], by use of advanced biomolecular technology, have unambiguously proven that both SNAT and ASMT are localized in the mitochondria and that melatonin is exclusively synthesized in the matrix of mitochondria. In addition, they also found that the melatonin receptor 1 (MT1) is also located in the mitochondrial membrane. The cell-protective effects of melatonin probably occur at the level of the mitochondria. On the other hand, the cofactor for SNAT to synthesize N-acetylserotonin, acetyl-CoA, is synthesized in the mitochondrial matrix at concentrations of 0.5–1.0 mM, while cytosolic acetyl-CoA only attains 3–30 μM. The calculated K_m_ of SNAT for acetyl-CoA is 0.11 ± 0.02 mM. Thus, the cytosolic acetyl-CoA level is far below the K_m_ of SNAT and only the mitochondrial concentration of acetyl-CoA is sufficient for exceeding the K_m_ of SNAT [44]. S-adenosylmethionine (SAM), the cofactor of ASMT, also indicates mitochondrial melatonin formation. SAM features a stable concentration in mitochondria compared to other cellular compartments [45] and fits, in this compartment, the catalytic K_m_ of ASMT. Therefore, an abundance of substrates and cofactors are in favor of mitochondrial melatonin synthesis. Melatonin can also be synthesized in extramitochondrial sites. For example, erythrocytes which are devoid of mitochondria have been reported to synthesize melatonin [46]. However, extramitochondrial melatonin synthesis is not as efficient as that in mitochondria. Thus, it may not be of biological significance. Collectively, mitochondrial intactness and functionality determine the melatonin production in organisms [47].

## 4. Potential Gender Bias in the Expression of *ASMT* in Mammals

As mentioned above, four enzymes are involved in mammalian melatonin synthetic pathway (Figure 1). Three of the genes are located in the autosomes, namely, *TPH*, *AADC* and *SNAT*. Thus, their expressions are without gender differentiations. In addition, isoenzymes exist for TPH and AADC and a mutation in one isoenzyme gene will not substantially impact the melatonin synthesis as mentioned above. However, the *ASMT* gene is located in the sex chromosomes, notably in both X and Y. Particularly, it localizes in the pseudoautosomal region 1 (PAR1) present in both sex chromosomes [48]. Additionally, a homologous sequence with a weaker binding of the respective fluorescence in situ hybridization (FISH) probe was detected in the subtelomeral region of the murine chromosome 9 [49]. Whether this sequence is functionally active remains uncertain. No isoenzymes of ASMT have been identified yet in mammals. However, near downstream of the *ASMT* sequence, another interesting gene, referred to as the *ASMT*-like gene (*ASMTL*), has been identified. This gene is a recombination of the full-length *ASMT* sequence (a homology of *ASMT*) connected with one of two bacterial genes of *maf*/*orfE* (maf: filamentation protein of *Bacilus subtilis*; orfE: orfE of Escherichia coli), respectively [49]. A combination of mammalian *ASMT* gene with bacterial genes may imply the evolutionary convergence of the melatonin synthetic pathway. Currently, the biological significance of the *ASMTL* is unknown but it highly expresses in all tissues tested, including spleen, thymus, prostate, ovary, testis, pancreas, placenta, intestine, colon, fibroblast, and fetal brain tissues. All these tissues feature the capacity to synthesize melatonin. Structural analysis shows that, similar to the ASMT, ASMTL has also conserved the catalytic and substrate binding domains for melatonin synthesis [49]. These indicate that ASMTL may directly or indirectly participate in melatonin production. The genes in the PAR1 of Y chromosome are similar to the genes in the autosomes and have their alleles in the respective sites of X chromosome. Whether this is also the case for *ASMT*, in terms of functional activity, remains to be clarified. Generally, all characterized genes within PAR1 escape X inactivation [50,51]; therefore, they are candidates for exerting gene dosage effects in sex chromosome aneuploidy conditions, such as Turner syndrome (45, X) or triple-X in women. Theoretically, the expressions of *ASMT* or *ASMTL* should not feature a gender bias as to the gene doses, since PAR genes are exempt from sex chromosome-specific heterochromatization. However, the genes located in the PAR1 of Y chromosome are subject to frequent recombination, shuffling, insertion, and mutation [52]. A crossover in the PAR is essential for the proper disjunction of X and Y chromosomes in male meiosis. An exceptionally high male crossover rate (17 fold higher than the genome-wide average) in humans has been reported [53]. This characteristic of PAR1 in Y chromosome inevitably leads to more frequent errors in genes of this part than in those of other chromosomes. High mutation rates in Y chromosomal *ASMT* are well documented in both humans [48] and mice [54]. In the murine pineal gland, they may contribute to the appearance of strains with relative melatonin deficiency. In humans, these *ASMT* mutations seem to associate with mental diseases [55,56]. As a result, the prevalence of the altered *ASMT* or *ASMTL* (assuming that the latter gene is also involved in melatonin synthesis) in males is higher than in the females, and this may result in the high frequency of dysfunctional *ASMT* or *ASMTL* genes that predominantly occur in males. Currently, there are no data to directly prove this. The observations from patients with autism spectrum disorder (ASD) may provide some clues. A lower-than-normal melatonin level, as well as a disrupted rhythm, have been reported in individuals with ASD [57]. This low melatonin in ASD is beyond the pineal gland’s size but may associate with deficiency in the melatonin synthetic pathway [58]. The mutations of *ASMT* are directly responsible for the low melatonin level in ASD [59]. A mechanistic study indicates that polymorphisms located in promoters of *ASMT*, including rs4446909 and rs5989681, cause a dramatic decrease in *ASMT* transcripts, ASMT activity, and melatonin levels in individuals with ASD [60]. These results suggest that a low melatonin level, caused by a primary deficit in ASMT activity, is a risk factor for ASD. The assumed higher chance of mutation in the Y chromosome would also imply that males should have a higher incidence of this disease than females. Indeed, ASD features an obvious dimorphism, with a male-to-female ratio of approximately 3 to 1 [61]. If we assume that *ASMT* polymorphisms are also present in the general population, but that these mutations are too mild to impair baseline melatonin synthesis, this implies that these hidden polymorphisms do not become phenotypically manifest in terms of ASD symptomatology. However, it may still be possible that the consequences of these mutations may become apparent as soon as the requirements for higher-than-baseline melatonin levels emerge under stressful conditions. Judging from the sexual distribution of patients with ASD, the male-to-female ratio in this mild *ASMT* mutation-related hidden polymorphism should also be male-dominated (around 3:1). Thus, the *ASMT*/*ASMTL* in PAR1 of the pseudoautosomale region may contribute to some degree to the potential dimorphism of melatonin production under certain conditions.

## 5. Potential Gender Bias of Mitochondria-Related Melatonin Synthesis

In organisms with sexual reproduction, mitochondria are basically inherited from females. The matrilineally related inheritance of mitochondrial genomes inevitably creates a sex-specific selective sieve. The selected traits seem unfavorable to males. For example, a germline mitochondrial mutation that is benign or only slightly deleterious in females may be harmful to males [62]. In addition, growing evidence shows that the male-harming mutations may exert pleiotropic beneficial effects on females and, thus, accumulate in the mitochondrial genome through positive selection [63,64]. These mutations are inherited through this unique asexually reproducing pathway of mitochondria. During evolution, such a mutation will accumulate in a population since this male-specific deleterious mtDNA mutation will not be subject to natural selection [65]. This sex-specific selective sieve on mitochondrial genomes may cause the innately functional deficiency of mitochondria in males compared to females. This phenomenon is referred to as mother’s curse [66,67]. On the other hand, there must be some ways to compensate this mother’s curse; otherwise, it would lead to species extinction. These compensations may include paternal leakage [68,69], interaction between mtDNA and nuclear genome [70], kin selection [71] and assortive mating [72]. However, these pathways cannot completely balance the mother’s curse and lead to some slightly compromised mitochondrial function in males. This male-related mitochondrial dysfunction may be linked to accelerated male aging [73] and a predisposition to viral infection, such as SARS-CoV-2 [74]. As mentioned above, mitochondria are the major sites of melatonin synthesis. The functional mitochondria are necessary for melatonin production, since the cofactors of acetyl-CoA and S-adenosylmethionine for SNAT and ASMT, respectively, are either the product of mitochondrial metabolism or concentrated in mitochondria. Any jeopardized mitochondrial function in males will directly impact their melatonin production to establish a dimorphism with a higher level in females. This dimorphism has only been observed under stressful, but not basal, physiological conditions. The reasons for this are discussed below.

## 6. Evidence to Support the Masked Dimorphism of Melatonin Production

The potential gender biases in the expression of ASMT/ASMTL in mitochondria that are in favor of females suggest a dimorphism in melatonin production, with higher levels in females than in males. This dimorphism is subtle under physiological conditions and is only manifested in aging and/or stressful conditions. For example, determinations of amplitude or the duration of nocturnal melatonin rises in healthy young lambs did not reveal any gender difference [75]. Similar results were also reported in a cohort of human subjects aged from 3 to 90 years old [76]. However, judging from the data available in this study, the majority of the subjects were distributed in the young-to-adult group, whereas only 8 subjects (1male and 7 females) in the group aged 80–90 were included. Thus, no meaningful information can be extracted concerning the sexual difference of melatonin production in the elderly population. On the other hand, a higher melatonin level in healthy young female subjects compared to age-matched males was also reported in another study; however, it remains doubtful whether this dimorphism in melatonin production is related to the oral contraceptive pill (OCP) of sex steroids used by females [77]. This suspicion is not supported by a previous study, which confirmed that sex steroids seem not to alter melatonin secretion in humans [78]. In fact, dimorphism in melatonin production has been observed in the elderly population. In a considerable large-scale clinical study involving 757 unselected elderly subjects aged 80.9 ± 9.7 years, a sex-difference with significantly higher levels of plasma melatonin in women than in men was demonstrated [79]. As an antioxidant and anti-inflammatory molecule [80,81], melatonin is stress-inducible. This has been proven in a variety of species, from unicellular organisms to mammals [82,83]. Lipopolysaccharide (LPS) injection is a common stressful, inflammation-inducing endotoxemic factor that can elicit raised melatonin production. When LPS was given to the Indian Palm squirrel (*F. pennanti*), a photoperiodic animal, no matter which photoperiodic regimens they are exposed to, the females always demonstrated a significantly greater serum melatonin concentration than the males, along with an increased survival rate [84]. In rats with experimental chronic kidney disease (CKD), females also exhibited significantly higher serum melatonin levels than males [85]. Hemorrhagic shock, another stressful stimulus, immediately induced a rise in melatonin in both male and female C3H/HeN mice. However, males returned to control level after two hours of hemorrhage, whereas a significantly increased plasma melatonin level was maintained in females after two hours of hemorrhage. This result also suggests that females have a long-lasting high melatonin synthetic capacity compared to the males. Another example concerns the melatonin levels in pregnant women. Pregnancy is a source of tremendous physiological stress for females. The level of melatonin is gradually increased in parallel with the progress of the pregnancy and reaches its peak at the full term [86]. This increase in melatonin production is believed to be beneficial for both mother and fetus. All the data mentioned above strongly suggest the presence of a subtle dimorphism in melatonin production across species, favoring the females. More well-designed studies are required to confirm this preliminary conclusion.

## 7. Discussion

Melatonin is a pleiotropic molecule and plays a variety of biological roles in animals. Its deficiency is associated with many aging-associated disorders, including neurodegenerative diseases, cancer, osteoporosis, diabetes, obesity, hypertension, susceptibility to infectious diseases, and autoimmunity [87,88,89,90,91,92,93]. Aging is a common factor related to melatonin deficiency. The underlying mechanisms are still not fully clarified. Since melatonin is synthesized in mitochondria, we hypothesize that compromised mitochondrial function with aging may contribute to aging-associated melatonin decline. Mitochondrial genomes are only transmitted via the matrilineal pathway. This allows females to develop and maintain superior mitochondrial function to males. If this is translated to melatonin production, it suggests that females produce higher levels of melatonin than males. However, this assumed dimorphism in melatonin production has not yet been clearly observed or fully clarified. To explain this seemingly controversial phenomenon, we need to introduce a new concept, i.e., the reserve, or maximum capacity, of melatonin synthetic function. This concept is, in formal terms, similar to that of the reserve capacity of heart function. For instance, in the resting condition, cardiac output is around 70 mL/stroke in a 70 kg body subject [94], which is referred to as the basal capacity of heart function. This basal capacity can increase as much as 4- to 6-fold during strenuous exertion [95]. The latter is referred to as the reserve, or maximum capacity, of heart function. Clearly, the huge difference between the basal and maximum capacity of heart functions is reserved for when it is needed. Similarly, we believe that there should also be basal and maximum capacities of melatonin synthesis, in a formal similarity to cardiac function. Under physiological conditions, cells may not need their maximum melatonin production; therefore, we refer to this as the basal capacity of melatonin synthetic function, notwithstanding the fact that pineal-derived melatonin already undergoes a considerable circadian variation. However, this kind of rhythmic dynamics does not contradict to the concept of basal vs. maximum capacity. The cardiac paradigm is also compatible with substantial circadian rhythms in heart function [96] and is, nevertheless, adaptable to strong exercise. Likewise, under stressful conditions, high levels of melatonin are required for its protective effects. The ability of the mitochondria to maximize their melatonin synthesis is referred to as the reserve/maximum capacity of melatonin’s synthetic function. The reserve/maximum capacity for melatonin synthesis in females is greater than that in males, accordingly based on the mitochondrial function. This indicates the presence of a potentially hidden dimorphism in melatonin production, favoring females. Under basal physiological conditions, this dimorphism is rather subtle and can only be detected as soon as the requirement of cells for melatonin production demands the maximum synthetic capacity of mitochondria. Males and females possess the capacity to adapt melatonin formation to the needs of the organism. They may generate basal levels of melatonin; thus, the low amounts of melatonin synthesized at basal capacity mask the potential dimorphism of melatonin production.

During aging, mitochondrial function gradually declines. The rate of decline in males is faster than that of females, perhaps due to the sexual bias of mitochondrial genomes. At the advanced age, such as over 70 years old with deleterious mitochondrial function, even under physiological conditions, the mitochondria in males can be incapable of synthesizing sufficient amounts of melatonin to match the requirements of cells. By contrast, the mitochondria in females may still possess the capacity to generate more melatonin than those in males. As a result, the dimorphism of melatonin production is manifested in the elderly population [79]. Melatonin is a potent mitochondrial protector. It reduces the oxidative stress of mitochondria and preserves mitochondrial function for ATP production [97,98]. In advanced age, compromised mitochondrial function jeopardizes melatonin production and leads to age-related melatonin deficiency. In turn, this further accelerates mitochondrial damage and results in a vicious cycle that is more obvious in males than in females. This phenomenon can explain the accelerated aging rate in elderly males compared to elderly females [99], especially with regard to the longevity bias of males and females, since melatonin and mitochondrial health both positively influence the aging process [100,101]. In this context, it should be noted that the evidence for both the sexual dimorphism in melatonin’s synthetic capacity and the age-related decline of melatonin are to date almost exclusively based on measurements of circulating and pineal melatonin. A problem of utmost importance is that of the differences and changes in extrapineal tissues. These studies are still in their infancy. There is no reason to conclude, given our current knowledge, that extrapineal tissue melatonin behaves in a similar way to that produced in and released by the pineal gland. A recent paper on the expression of SNAT/AANAT and ASMT in the murine small intestine and colon did not reveal a steady decline [102]. Instead, their expression rates demonstrated complex patterns that comprised considerable increases in the course of aging. We may take this as a hint for the existence of tissue responses that lead to increased melatonin formation in extrapineal tissues under conditions of need and stress. Accordingly, tissues may respond, for the purpose of self-protection, to stressful conditions, including inflammation, hypoxia, tissue ischemia/reperfusion, metabolic disorders and diseases by increasing melatonin production as high as possible [103]. This inducible trait of melatonin production allows to approach the reserve/maximum capacity of melatonin’s synthetic function in organisms. Due to the potential sex bias of *ASMT*/*ASMTL* expressions and/or sex bias of mitochondrial function, females should have a larger reserve/maximum capacity of synthetic melatonin function compared to males. Therefore, females synthesize more melatonin than males if required and, thus, the dimorphism in melatonin production is measurable under stressful conditions. Under normal conditions, the high reserve/maximum capacity of melatonin’s synthetic function in females is masked by low melatonin demand, since the basal capacities of both sexes are sufficient for satisfying the tissue’s basal requirement. This does not mean that it is not important. The high reserve/maximum capacity of melatonin’s synthetic function creates a better level of fitness by being prepared for responses to unpredicted internal and external environmental changes. The biological significance of this high reserve/maximum capacity is also to improve the recovery rate of organisms from injury, to increase resistance to pathogen infection, and even to enhance their survival chances by maximizing their melatonin production under stressful conditions. These advantages have been demonstrated in transgenic melatonin-enriched animals. *SNAT* or *ASMT* overexpression in sheep improves their reserve/maximum capacity of synthetic melatonin function compared to WT. As a result, these transgenic sheep exhibit a high tolerance to LPS challenge and also display enhanced resistance to the infectious disease, brucellosis, compared to WT [104,105]. Considering the higher reserve capacity for melatonin synthesis in women compared to in men, women seem to possess a significant advantage over men in tolerating stressful conditions, such as those of infectious diseases [106]. A current example is the obvious gender difference in SARS-CoV-2 infection. Female COVID-19 patients experience, on average, less serious symptoms and lower death rates than males, findings that have been attributed to a variety of factors [107,108]. The disadvantages identified in male COVID19 patients have also been linked to the deficiencies of mitochondrial function and melatonin production [109,110,111]. To support this argument, melatonin treatment has reduced the severity of symptoms and promoted the recovery of patients with COVID-19 [112]. Most importantly, melatonin administration significantly reduced the mortality of severe COVID-19 patients by proximately 93% in a recent clinical trial [113]. Macrophages and lymphocytes being important immune cells are involved in the innate and adaptive immunities in infectious diseases, including COVID-19 infection. Both of these cell types also feature the capacity to synthesize melatonin [114,115]. Whether there is a dimorphism in melatonin production during pathogen infection, and whether this potential dimorphism impacts disease progression differently in the two sexes, are interesting questions that require further studies for clarification.

We also should note that the deleterious rate of mitochondrial function in different tissues varies with aging [116]. This should be mirrored by melatonin’s synthetic capacity. Moreover, the dimorphism of melatonin production may vary with aging and among tissues. This issue should be carefully considered in the design of future studies to investigate the potential dimorphism in melatonin production, especially in specific tissues.

In conclusion, based on sex-differentiated genetic traits and published data, evidence strongly supports the presence of a dimorphism in melatonin production in favor of females. This dimorphism is subtle and it is masked under normal conditions; however, it is manifested in advanced age or in response to stressful stimuli. This phenomenon can be explained by the reserve/maximum capacity of melatonin’s synthetic function. This dimorphism in melatonin production may relate to the longevity dimorphism between males and females and also to their different tolerances to stressors.

## Figures and Tables

**Figure 1 molecules-26-07302-f001:**
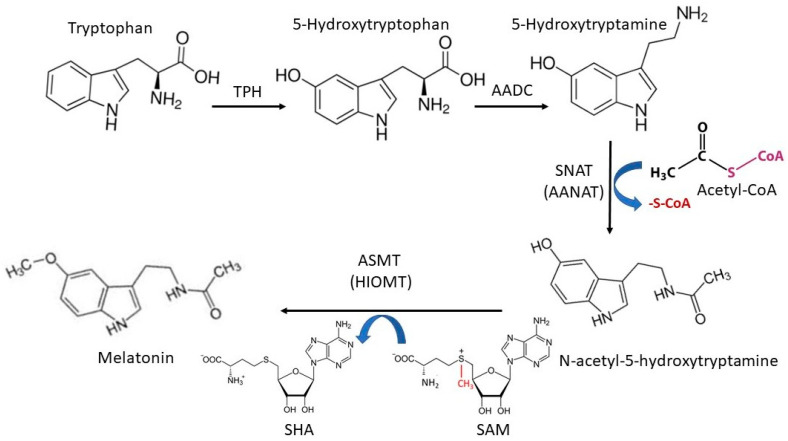
Melatonin synthetic pathway in animals.

## Data Availability

Not Applicable.

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
