# Peer review of "The Reserve/Maximum Capacity of Melatonin’s Synthetic Function for the Potential Dimorphism of Melatonin Production and Its Biological Significance in Mammals"

_molecules, 2021, doi:10.3390/molecules26237302_

Round 1

Reviewer 1 Report

The paper by Tan and Hardeland is a very timely and comprehensive review article. Authors compiled the state-of-the-art involved in the mechanisms whereby the production of melatonin is elevated in females considering local synthesis facing threating or stressful conditions. Authors introduce a new concept for the melatonin synthesis, termed “reserved or maximum capacity of melatonin” referring to its maximal particular dimorphic production under complex organismal conditions. These concentrations are expected to be optimal in females than male individuals.

I have only few suggestions to the manuscript

Define first AFMK and AMK before abbreviation.

Page 5, Please add reference to “Structural analysis shows that similar to the ASMT, ASMTL also has the conserved catalytic and substrate binding domains for melatonin synthesis”.

Interestingly, in female tumors (especially ovarian and cervical cancers), the melatonergic indices are low and predict worse prognosis. Do authors took into considerations the genes of CYP’s family associated with the metabolism of melatonin. Is the time of melatonin metabolism/transformation expected to be the same for male and females’ tissues and cells? 

The capacity of women potentially producing high melatonin levels can also be found during gestation. Perhaps, the ability to modify its production to preserve the oocytes’ quality and, subsequently, the gestational steps may render women in advantage. Whether this tissue-specific synthesis is dependent on ASMT/ASMTL transcripts needs further investigation. 

Author Response

Thanks for your comments and suggestions. Following is our responses to the comments and suggestions point by point. In the text, the changes were highlighted in red.

    Thanks.  

Reviewer 1.

Comments and Suggestions for Authors

The paper by Tan and Hardeland is a very timely and comprehensive review article. Authors compiled the state-of-the-art involved in the mechanisms whereby the production of melatonin is elevated in females considering local synthesis facing threating or stressful conditions. Authors introduce a new concept for the melatonin synthesis, termed “reserved or maximum capacity of melatonin” referring to its maximal particular dimorphic production under complex organismal conditions. These concentrations are expected to be optimal in females than male individuals.

I have only few suggestions to the manuscript

Define first AFMK and AMK before abbreviation.

Done

 Page 5, Please add reference to “Structural analysis shows that similar to the ASMT, ASMTL also has the conserved catalytic and substrate binding domains for melatonin synthesis”.

Done, reference added.

 Interestingly, in female tumors (especially ovarian and cervical cancers), the melatonergic indices are low and predict worse prognosis. Do authors took into considerations the genes of CYP’s family associated with the metabolism of melatonin. Is the time of melatonin metabolism/transformation expected to be the same for male and females’ tissues and cells?

This paper is concentrated on melatonin synthesis. The genes of the melatonin metabolizing enzymes of the CYP’s family are not located in the sex chromosomes. Probably, they have only a weak relationship with this dimorphism of melatonin production.

The capacity of women potentially producing high melatonin levels can also be found during gestation. Perhaps, the ability to modify its production to preserve the oocytes’ quality and, subsequently, the gestational steps may render women in advantage. Whether this tissue-specific synthesis is dependent on ASMT/ASMTL transcripts needs further investigation.

Thanks for your constructive suggestion. We have added this point in the section 6.

Reviewer 2 Report

[molecules-1479350]:  Although this is a good review, some comments and suggestions follow for the authors.

For this reviewer, the objective is not sufficiently clear.

Line 140: Plese define “SAM”

Lines 140-144: Remove “TPH: tryptophan hydroxylase, AADC: aromatic amino acid ecarboxylase, SNAT: serotonin N-acetyltransferase, AANAT: arylalkylamine N-acetyltransferase, ASMT: N-acetylserotonin-O-methyltransferase, HIOMT: hydroxyindole-O-methyltransferase, SAM: S-adenosylmethionine, SAH: S-Adenosylhomocysteine.”

Figure 1: “SHA” is not described in the text.

Lines 180-182: “The genes of three of the four enzymes involved in mammalian melatonin synthetic pathway are located in the autosomes, namely, TPH, AADC and SNAT.”

Line 187: Please define “FISH”

Line 208: Please define “PARP”

Line 245: basically or only?

Line 270: We recommend, to provide a brief discussion from the described data.

Line 295: “Funambulus pennanti”

Lines 302-303: “...after this period”

Line 310: Remove ”Its production wanes with aging.”

At the end of the text, it not clear what of conclusions are arising from the described data.

Finally, an English editing is highly recommended. Furthermore, we recommend more care in formatting the manuscript, including references list.

Author Response

Thanks for your comments and suggestions. Following is our responses to the comments and suggestions point by point. In the text, the changes were highlighted in red.

    Thanks.  

Reviewer 3 Report

The manuscript describes an interesting and exciting hypothesis. Of special interest is the possible role of  melatonin concerning the gender difference in the immune response (Kloc M, Ghobrial RM, Kubiak JZ, 2020) also in relation of the  emerging pineal-immune system axis and its role in the immune response.  Although the authors briefly quote the  importance of immunity in combating diseases and hence stressfull situations, I think this issue deserves more attention and could also provide an experimental approach to challenge the authors hypothesis. In fact, antigen-stimulated macropahges can synthesize melatonin to equilibrate the ongoing immune response, and to my knowledge no data are available about a possible sex difference in this ability. References are listed in a rather fancy style.

Author Response

(The authors gave the same response as above.)
